# Acknowledging the impact of seasonal blood pressure variation in hypertensive CKD and non-CKD patients living in a Mediterranean climate

Tatiana Charbel[1], Georgio El Koubayati[2], Chloe Kharsa[1], Mabel Aoun[3,4]*

1 Faculty of Medicine, Department of Internal Medicine, Saint-Joseph University, Beirut, Lebanon, 2 Faculty of Medicine, Department of Internal Medicine, Lebanese University, Beirut, Lebanon, 3 Faculty of Medicine, Department of Nephrology, Saint-Joseph University, Beirut, Lebanon, 4 AUB Santé, Lorient, France

* aounmabel@yahoo.fr

## Abstract

### Background

This study aims to assess seasonal blood pressure (BP) variation in chronic kidney disease (CKD) and non-CKD patients living in a Mediterranean climate, and to find out if this variation entails significant adjustment of treatment and if it impacts renal outcomes and mortality.

### Methods

This retrospective study included all hypertensive patients seen between February 2006 and April 2020 in two Lebanese clinics. Regression analyses were used to assess the association of seasonal BP variability and treatment adjustment with eGFR change from baseline, dialysis initiation and death.

### Results

A total of 398 patients of 64.2 ±13.9 years were followed for 51.1 ±44.3 months, 67% had eGFR< 60 mL/min. Mean systolic and diastolic BP was 137.7 ±14.7 and 76.5 ±9.5 mmHg respectively. Systolic and diastolic BP were significantly lower in the warm season in CKD and non-CKD patients ($P$<0.001). The majority (91.4%) needed seasonal treatment modifications. After adjustment to age, sex, baseline eGFR, BP and number of antihypertensive drugs, we found a significant loss of eGFR with treatment modifications in both seasons, double risk of dialysis with the increase of antihypertensive treatment in both seasons and a 2.5 more risk of death with reduced treatment in the warm season.

### Conclusion

This study confirmed the seasonal BP variability in CKD and non-CKD patients from a Mediterranean climate. All types of treatment adjustment were associated with eGFR loss. Low BP in the warm season was highly associated with death.

**Data Availability Statement:** All relevant data are within the paper and its Supporting Information files.

**Funding:** The authors received no specific funding for this work.

**Competing interests:** The authors have declared that no competing interests exist.

# Introduction

High blood pressure (BP) is an important risk factor for cardiovascular and kidney diseases. Due to its high prevalence, with 1.28 billion adults being hypertensive worldwide, it has become a global health issue and requires the development of new strategies for the prevention of non-communicable diseases [1,2]. Hypertension is more common in patients with chronic kidney disease (CKD) [3,4]; its prevalence ranges between 35.8% to 84.1% in CKD patients versus 23.3% in patients without CKD [5]. CKD can both be the cause and the consequence of hypertension. The mechanisms of hypertension in CKD include volume overload, sympathetic over-activity, salt retention, endothelial dysfunction, and alterations in hormonal systems that regulate BP [6]. Likewise, hypertension remains a leading attributed cause of end-stage kidney disease [7], as well as a major risk factor for cardiovascular morbidity and mortality [3,8]. Recommendations on BP management were recently harmonized [9]. According to the American College of Cardiology/American Heart Association and European Society of Cardiology/European Society of Hypertension guidelines, the target systolic blood pressure (SBP)/diastolic blood pressure (DBP) is <130/80 mmHg as a general treatment target for all hypertensive patients in order to reduce morbidity and mortality, provided that BP was accurately measured [9,10].

Beyond and independent of elevated mean BP, studies have shown that both short-term and long-term increases in blood pressure variability (BPV) are associated with an increased incidence of cardiovascular events and mortality [5,11–13]. Within-patient BPV between visits to a physician's office has been proven strongly prognostic for cardiovascular morbidity [13,14]. One of the reported causes of BPV is seasonal climatic changes [15–18], where systolic and diastolic BP have been reported to be lower during summer and higher during winter, mainly as a result of changes in outdoor temperature [19]. In a Chinese study of 506 673 adults of the general population, the variation in systolic blood pressure between summer and winter season averaged 10 mmHg [20]. When it comes to the risk of cardiovascular events, the role of climatic changes has also been proven to increase heat- and cold-related mortality due to a multifactorial mechanism where BP plays an important role [21,22].

Several studies assessed the impact of visit-to-visit or day-to day BPV in CKD patients on kidney function, dementia and mortality [23–27] but to our knowledge very few evaluated the seasonal variation of BP in CKD patients [28]. This study aims to find out whether there is a seasonal variation of blood pressure in CKD and non-CKD patients living in a Mediterranean climate, whether this variation entails frequent adjustment of treatment, and finally whether it impacts renal and survival outcomes.

# Materials and methods

## Study design and participants

This is a retrospective study that included all patients with hypertension seen for the first time between February 2006 and April 2020 in two private Lebanese nephrology clinics. These two clinics are situated at two different altitude levels (sea level and 800 meters) and are both located in the Mount Lebanon region [29]. This region has a Mediterranean, hot summer climate (Csa subgroup, according to the Köppen–Geiger climate classification system) [30]. Its average yearly temperature is 21.21˚C (70.18˚F) and it is 1.29% higher than Lebanon's averages [31], which we collected from January 2006 to June 2022 (S1 Table). Theoretically, four seasons exist in Lebanon: autumn (October-December), winter (December-March), spring (March-May) and summer (June-September). For the sake of this study, variations of temperature were analyzed based on two seasons: the warm season that includes the warmest months

from May to October and the cold season that includes the coldest months from November to April (S1 Table).

## Blood pressure measurement and management

Blood pressure was measured in the seated position, after at least 10 minutes of rest, in a quiet environment at the two clinics, using an automated arm blood pressure monitor. In the majority of cases, levels at both arms were recorded with three consecutive measurements. The third reading was collected. The management of hypertension in these clinics followed the international guidelines, antihypertensive treatment consisted of a renin-angiotensin-aldosterone system (RAAS) inhibitor at initiation of therapy combined with either a calcium channel blocker (CCB) or a thiazide diuretic if needed. A serum creatinine that increased above 30% of baseline led to a drop of RAAS inhibitors. Angiotensin receptor blockers (ARBs) were the preferred therapy for CKD patients and angiotensin converting enzyme (ACE) inhibitors were used in patients with coronary artery disease or heart failure. Moxonidine and alpha-blockers were used as an add-on therapy in severe cases of hypertension, especially when RAAS inhibitors were no longer tolerated because of hyperkalemia in advanced stages of CKD or acute increase in serum creatinine.

## Sample size

The minimum sample size was calculated using an online sample size calculator. We assumed an effect size of 0.5, an alpha error of 5%, and a power of 95% which leads to a representative sample of at least 384 patients.

## Eligibility criteria

Subjects were included if they were older than 18 years, with or without CKD (defined as per the KDIGO 2012 clinical practice guidelines [32]), diagnosed and/or followed up between February 2006 and April 2020, hypertensive with hypertension defined as being treated with antihypertensive medications or as SBP/DBP >130/80 mmHg as per the 2017 updated American College of Cardiology/American Heart Association guidelines [33], with at least 2 visits to the clinic in which BP was measured. Patients were excluded if there was substantial missing data in their medical files and if all their visits happened within the same month or two of each year.

## Variables collected

Data collection through chart review started in August 2022 and ended in February 2023. The variables were collected from medical files of patients and included age, sex, smoking status, number of visits, duration of follow-up, BMI, diabetes, coronary artery disease (CAD), systolic heart failure, SBP and DBP levels of all visits (if there was a modification of treatment, SBP and DBP levels before any modification were collected), laboratory measurements of serum creatinine and albumin over creatinine ratio (ACR) at first visit T1 and last visit T2, mean monthly outdoor temperature during each visit based on S1 Table. Antihypertensive treatment type was recorded: CCB, ARB, ACE inhibitor, beta-blocker, thiazide, furosemide, spironolactone, moxonidine, alpha-blocker. All changes in antihypertensive medications were collected: dose reduction or increase, removing or adding a medication, change of medication.

## Definitions

CKD stages were defined based on the 2012 KDIGO classification of CKD that takes into account the glomerular filtration rate estimated from the CKD-EPI equation [33]. Patients

with cystic diseases, solitary kidney or glomerular diseases with normal eGFR and ACR levels were considered as CKD patients. Pulse pressure was defined as the SBP minus the DBP. The number of antihypertensive treatments was computed by adding all antihypertensive medications prescribed to the patient during the entire follow-up (ACE inhibitors and ARB were considered as the same drug class under RAASi).

## Collection and averaging of BP levels

SBP and DBP levels were collected for every patient from all their visits across all seasons, before any treatment adjustments were made during each visit. These SBP and DBP levels were grouped and averaged for each cold or warm season, for each patient.

## Ethical considerations

This study was conducted in accordance with the Declaration of Helsinki 1975. It received the approval of the HDF/Saint Joseph University Ethics Committee (CEHDF 2045). The requirement for informed consent was waived by the ethics committee due to the retrospective design of the study.

## Statistical analysis

The Statistical Package for the Social Sciences (SPSS) version 25 was used for data analysis. Categorical variables are presented as numbers and percentages. Continuous data is reported as mean and standard deviation (SD) if it has a normal distribution, and as median and inter quartile range (IQR) if the data is not normally distributed. A paired-T test analysis was used to assess the variation between summer and winter blood pressure values. McNemar's test was used to compare paired categorical variables. A linear regression analysis was used to assess the association between seasonal variability and the change in eGFR. A logistic regression analysis was used to assess the relationship between seasonal variability and dialysis or death. Variables that had a significant $p$-value in univariate analysis were adjusted to demographics, eGFR, SBP, DBP, and number of antihypertensive treatments when included in the multivariable analysis. We had missing data for serum creatinine at T2 for 17 patients. These patients were not included in the linear regression that analyzed the outcome "eGFR change from baseline". $P$-value was considered significant if less than 0.05.

# Results

## General characteristics of patients

About 1400 medical files were reviewed and 398 patients were selected for inclusion in the final analysis (S1 Fig). Among these 398 patients, 344 patients had CKD based on KDIGO definition. Their mean age was 64.2 ±13.9 years and 65.3% of them were males. Their mean follow-up was 51.1 ±44.3 months (49.9±42.5 months in CKD patients and 57.2 ±53 months in non-CKD patients). Two-hundred and sixty-eight patients (68%) had an eGFR< 60 mL/min/1.73 m$^2$. The mean number of antihypertensive drugs was 3.4 ±1.5. Their mean systolic and diastolic BP during the entire follow-up was 137.7 ±14.7 and 76.5 ±9.5 mmHg respectively. Table 1 summarizes the characteristics of these patients.

## Seasonal variation of temperature, blood pressure and heart rate

The systolic and diastolic BP were both significantly lower during the warm season (Table 2). There was no difference in heart rate between seasons.

**Table 1. General characteristics of patients.**

| Variable | Total number N = 398 |
|---|---|
| **Baseline demographics, CV risk factors and laboratory values** | |
| **Age, years, Mean ±SD** | 64.2 ±13.9 |
| **Sex, F/M, n (%)** | 138/260 (34.7/65.3) |
| **Diabetes, n (%)** | 190 (47.7) |
| **Obesity, n (%)** | 104 (26.1) |
| **CAD, n (%)** | 128 (32.2) |
| **Heart failure, n (%)** | 52 (13.1) |
| **Average systolic blood pressure of entire follow-up, mmHg, Mean ±SD** | 137.7 ±14.7 |
| **Average diastolic blood pressure of entire follow-up, mmHg, Mean ±SD** | 76.5 ±9.5 |
| **Average pulse pressure of entire follow-up, mmHg, Mean ±SD** | 61.2 ±14.8 |
| **Serum creatinine at T1, mg/dL, Median [IQR]** | 1.5 [1,2.1] |
| **eGFR at T1, mL/min/1.73 m2, Mean ±SD** | 51.7 ±30.3 |
| **Albumin to creatinine ratio at T1, mg/g, Median [IQR]** | 202 [34.5,1049.3] |
| **CKD, n (%)** | 344 (86.4) |
| **CKD stages, n (%)** Stage 1 Stage 2 Stage 3a Stage 3b Stage 4 Stage 5 | 39 (11.3) 38 (11) 60 (17.4) 101 (29.4) 80 (23.3) 26 (7.5) |
| **Causes of CKD, n (%)** Diabetes Cystic disease Glomerular disease Solitary kidney Other | 190 (55.2) 19 (5.5) 54 (15.7) 18 (5.2) 63 (18.3) |
| **Medications** | |
| **ACE inhibitors, n (%)** | 98 (24.6) |
| **Angiotensin Receptor Blockers, n (%)** | 282 (70.9) |
| **RAAS inhibitors, n (%)** | 312 (78.4) |
| **Calcium channel blockers, n (%)** | 317 (79.6) |
| **Alpha-blockers, n (%)** | 94 (23.6) |
| **Beta-blockers, n (%)** | 289 (72.6) |
| **Thiazides, n (%)** | 143 (35.9) |
| **Furosemide, n (%)** | 158 (39.7) |
| **Moxonidine, n (%)** | 99 (24.9) |
| **Number of antihypertensive treatments, Mean ±SD** | 3.3 ±1.4 |
| **Follow-up and outcomes** | |
| **Follow-up duration in months, Mean ±SD** | 51.1 ±44.3 |
| **Number of visits, Median [IQR]** | 5 [3,10] |
| **Number of visits over follow-up duration in months, Median [IQR]** | 0.2 [0.1,0.3] |
| **Serum creatinine at T2, mg/dL, Median [IQR]** | 1.6 [1,2.9] |
| **eGFR at T2, mL/min/1.73 m2, Mean ±SD** | 46.0 ±32.1 |
| **Albumin to creatinine ratio at T2, mg/g, Median [IQR]** | 242 [30,982.5] |
| **eGFR change from baseline, Mean ±SD** | -4.1 ±17.4 |
| **eGFR change from baseline per year, Median [IQR]** | -1.3 [-3.6,1.2] |
| **Dialysis, n (%)** | 80 (20.1) |
| **Death, n (%)** | 57 (14.3) |

**Table 2. Comparison of blood pressure levels between two seasons.**

| Variable | Cold season | Warm season | P |
|---|---|---|---|
| Weather temperature, ˚C | 11.5 ±2.1 | 22.4 ±2.4 | <0.001 |
| Systolic Blood Pressure, mmHg | 140.9 ±15.9 | 133.9 ±15.4 | <0.001 |
| Diastolic Blood Pressure, mmHg | 77.4 ±10.3 | 74.3 ±9.9 | <0.001 |
| Pulse pressure, mmHg | 62.8 ±15.7 | 59.8 ±15.9 | <0.001 |
| Heart Rate | 72.7 ±12.6 | 72.9 ±12.8 | 0.801 |

Note. Data is reported as mean ±SD; Paired t-test was used to compare the two groups; This study considered the warm season as the one from May to October and the cold season from November to April.

When patients were divided into CKD and non-CKD, systolic and diastolic BP remained significantly lower in the warm season in both groups, whether CKD was defined as having albuminuria/structural anomalies or eGFR<60 mL/min/1.73 m$^2$ or whether considered as having renal failure with eGFR<60 mL/min/1.73 m$^2$ (Table 3).

## Seasonal adjustment of antihypertensive treatment

A total of 362 patients (91.4%) needed antihypertensive treatment modification during their follow-up (Table 4).

The reduction of dose and removal of a medication were significantly higher in warm seasons (Table 4). Among the 82 reduced medications in warm seasons, 32 were RAASi (39%), 27 CCBs (32.9%), 21 diuretics (25.6%), 19 beta-blockers (23.2%). Among the removed 122 medications in warm seasons: 53 were RAASi (43.4%), 36 diuretics (29.5%), 34 CCB (27.9%).

The increase in dose and adding a medication were significantly higher in cold seasons (Table 4). Among the 131 increased medications, 53 (40.5%) were CCBs, 41 (31.3%) diuretics, 33 (25.2%) RAASi, 7 (5.3%) beta-blockers, 5 (3.8%) alpha-blockers and 4 (3%) moxonidine. Among the added 182 medications during cold season: 62 were RAASi (34.1%), 46 diuretics (25.3%), 46 CCBs (25.3%), 16 beta-blockers (8.8%), 15 moxonidine (8.2%), 13 alpha-blockers (7.1%).

**Table 3. Comparison of patients with CKD vs non-CKD.**

| Variable | Cold season | Warm season | P | Cold season | Warm season | P |
|---|---|---|---|---|---|---|
| | CKD, n = 344 | | | Non-CKD, n = 54 | | |
| Weather temperature, ˚C | 11.5 ±2.1 | 22.5 ±2.3 | <0.001 | 11.5 ±2.3 | 21.8 ±3.2 | <0.001 |
| Systolic Blood Pressure, mmHg | 140.9±16.3 | 134.2 ±15.2 | <0.001 | 140.9 ±14.1 | 131.9 ±17.0 | 0.007 |
| Diastolic Blood Pressure, mmHg | 77.1 ±10.4 | 74.1 ±10.0 | <0.001 | 79.5 ±9.4 | 76.0 ±9.6 | 0.027 |
| Pulse pressure, mmHg | 63.2 ±15.9 | 60.3 ±16.1 | <0.001 | 60.2 ±13.8 | 55.9 ±14.3 | <0.001 |
| Heart Rate | 72.9 ±12.9 | 72.9 ±13.2 | 0.838 | 71.7 ±11.3 | 72.9 ±10.2 | 0.867 |
| | CKD stages 3 to 5, n = 268 | | | Non-CKD and CKD stages 1 to 2, n = 130 | | |
| Weather temperature, ˚C | 11.6 ±2.0 | 22.6 ±2.1 | <0.001 | 11.3 ±2.2 | 21.9 ±2.9 | <0.001 |
| Systolic Blood Pressure, mmHg | 142.4 ±16.3 | 135.6 ±15.2 | <0.001 | 137.5 ±14.8 | 129.9 ±15.1 | <0.001 |
| Diastolic Blood Pressure, mmHg | 76.1 ±10.6 | 73.1 ±10.0 | <0.001 | 80.4 ±8.9 | 77.1 ±9.4 | 0.001 |
| Pulse pressure, mmHg | 66.0 ±15.6 | 62.8 ±15.9 | <0.001 | 56.3 ±13.8 | 52.9 ±13.7 | <0.001 |
| Heart Rate | 71.7 ±12.1 | 71.8 ±12.3 | 0.836 | 75.3 ±13.6 | 75.5 ±13.9 | 0.884 |

Note. Data is reported as mean ±SD; Paired t-test was used to compare the two groups; This study considered the warm season as the one from May to October and the cold season from November to April.

**Table 4. Comparison of different treatment adjustment between the two seasons.**

|  | Warm season | Cold season |  |
| --- | --- | --- | --- |
| Variable | Total, N = 398 | Total, N = 398 | *P* |
| Reduction of dose | 82 (20.6) | 54 (13.6) | 0.025 |
| Removal of medication | 122 (30.7) | 69 (17.3) | <0.001 |
| Increase of dose | 88 (22.1) | 131 (37.1) | <0.001 |
| Adding medication | 155 (38.9) | 182 (45.7) | 0.040 |
| Change of medication | 146 (36.7) | 132 (33.2) | 0.442 |

Note. McNemar's test was used to compare the two groups. Change of medication is defined as replacing one medication by another without increasing or decreasing the number or doses of medication).

## Impact of BP seasonal variation and of treatment adjustment on renal outcomes

Two renal outcomes were assessed: the eGFR change from baseline (Table 5A and 5B) and reaching dialysis (Table 6A and 6B). A linear regression analysis assessed factors associated with eGFR change from baseline: after adjustment to age, sex, baseline eGFR and levels of systolic and diastolic blood pressure, we found that adding or removing a medication during warm season is associated with a statistically significant loss of eGFR of 6.5 and 5.7 ml/min respectively. An increase or a reduction in dose during the cold season are also associated with a significant loss of 7 and 6.6 mL/min of eGFR respectively.

Regarding the outcome of dialysis, the logistic regression analysis showed that one mmHg increase in systolic blood pressure during cold season increases the risk of dialysis by 2%. And an increase of antihypertensive treatment during warm or cold season multiplies the risk of dialysis by 2.

## Impact of seasonal BP variation on death

Age, eGFR and the pulse pressure are associated with an increased risk of death. After adjustment to age, sex, baseline eGFR, average systolic and diastolic BP, reduction of antihypertensive dose during the warm season multiplies the risk of death by 2.5 (Table 7A and 7B).

## Discussion

According to this study, seasonal blood pressure variation is significant in both CKD and non-CKD hypertensive patients. It showed that, in a Mediterranean climate with a mean difference of 10 degrees Celsius between the warm and cold season, the seasonal difference in SBP averaged 9 mmHg in non-CKD patients and 6 mmHg in CKD patients. These findings align with previous studies accumulating evidence that BP globally varies with climatic circumstances in all hypertensive patients, no matter the gender, the age, the geographic location or the kidney function [20,28,34,35]. This can be explained by physiological thermoregulation like temperature-induced vasoconstriction and vasodilation, climate-related perspiration, changes in dietary and water intake [36]. Our results point out to a change in hydration status knowing that 26% of patients stopped diuretics and 30% of them removed diuretics during the warm season. An additional reason behind the lower BP in warm seasons could be related to vitamin D [37]. Although interventional trials did not show a decrease in BP following vitamin D supplementation, several studies emphasized the relationship between exposure to sunshine and the release of nitric oxide from the skin [37]. To our knowledge, only one published study tackled seasonal BPV in patients with CKD, living in Taiwan [28]. Their sample included 508 CKD

**Table 5.  a. Univariate regression analysis of factors affecting eGFR change from baseline.** b. Adjusted linear regression analysis of factors affecting eGFR change from baseline.

| Variable | Unstandardized coefficient B | 95% Confidence Interval | P |
|---|---|---|---|
| Age in years | 0.096 | -0.032; 0.224 | 0.140 |
| Sex, *Ref*: *Male* | 2.137 | -1.537; 5.812 | 0.253 |
| eGFR at baseline in mL/min/1.73 m² | -0.071 | -0.131; -0.012 | 0.019 |
| eGFR< 60 mL/min/1.73 m² | 1.882 | -1.920; 5.684 | 0.331 |
| Difference in Average Systolic Blood Pressure between cold and warm season | -0.017 | -0.151; 0.117 | 0.799 |
| Difference in Average Diastolic Blood Pressure between cold and warm season | -0.108 | -0.324; 0.108 | 0.326 |
| Average Systolic Blood Pressure during cold season | -0.029 | -0.144; 0.085 | 0.617 |
| Average Diastolic Blood Pressure during cold season | -0.002 | -0.183; 0.178 | 0.980 |
| Average Pulse Pressure during cold season | -0.030 | -0.148; 0.088 | 0.618 |
| Average Systolic Blood Pressure during warm season | -0.008 | -0.127; 0.110 | 0.889 |
| Average Diastolic Blood Pressure during warm season | 0.038 | -0.145; 0.221 | 0.684 |
| Average Pulse Pressure during warm season | -0.024 | -0.141; 0.094 | 0.690 |
| Number of antihypertensive treatments | -1.254 | -2.662; -0.154 | 0.081 |
| ACE inhibitors | -2.302 | -6.322; 1.719 | 0.261 |
| Angiotensin receptor blockers | -4.250 | -8.092; -0.407 | 0.030 |
| Calcium channel blockers | -3.091 | -7.602; 1.419 | 0.179 |
| Alpha-blockers | -1.841 | -5.848; 2.165 | 0.367 |
| Beta-blockers | 1.932 | -2.035; 5.899 | 0.339 |
| Thiazide diuretics | 0.301 | -3.322; 3.924 | 0.870 |
| Moxonidine | -4.795 | -8.689; -0.901 | 0.016 |
| Reduction of dose during warm season | -2.300 | -6.731; 2.131 | 0.308 |
| Removal of medication during warm season | -4.078 | -7.972; -0.183 | 0.040 |
| Increase of dose during warm season | 0.532 | -3.805; 4.869 | 0.810 |
| Adding medication during warm season | -5.942 | -9.655; -2.229 | 0.002 |
| Change of medication during warm season | -1.079 | -6.155; -1.322 | 0.576 |
| Increase of dose during cold season | -6.689 | -10.543; -2.834 | 0.001 |
| Adding medication during cold season | -3.556 | -7.306; 0.194 | 0.063 |
| Reduction of dose during cold season | -6.239 | -11.394; -1.084 | 0.018 |
| Removal of medication during cold season | -4.689 | -9.403; 0.025 | 0.051 |
| Change of medication during cold season | -4.795 | -8.658; -0.932 | 0.015 |

| | Adjusted to age, sex and baseline eGFR | | | Adjusted to age, sex, baseline eGFR, number of antihypertensive treatments, average systolic and diastolic BP | | |
|---|---|---|---|---|---|---|
| Variable | Unstandardized coefficient B | 95% Confidence Interval | P | Unstandardized coefficient B | 95% Confidence Interval | P |
| Angiotensin receptor blockers | -3.719 | -7.631; 0.193 | 0.062 | | | |
| Moxonidine | -6.852 | -10.916; -2.787 | 0.001 | -4.085 | -9.269; 1.099 | 0.122 |
| Removal of medication during warm season | -4.731 | -8.649; -0.813 | 0.018 | -5.161 | -9.393; -0.929 | 0.017 |
| Adding medication during warm season | -6.154 | -9.866; -2.441 | 0.001 | -5.832 | -9.808; -1.856 | 0.004 |

(*Continued*)

| | | | | | | |
|---|---|---|---|---|---|---|
| **Increase of dose during cold season** | -7.045 | -10.890; -3.200 | <0.001 | -6.090 | -10.189; -1.991 | 0.001 |
| **Reduction of dose during cold season** | -6.616 | -11.751; -1.480 | 0.012 | -4.725 | -10.322; 0.873 | 0.098 |
| **Change of medication during cold season** | -4.617 | -8.492; -0.741 | 0.020 | -3.673 | -7.911;0.564 | 0.089 |

patients and most of them had an eGFR< 60 mL/min/1.73 m$^2$. They did not provide information about the difference in weather temperature between winter and summer and their analysis was different from our study for they divided patients into four groups based on the BP variation trend between the two seasons. They concluded that patients who had persistent BP elevation between summer and winter had worse outcomes [28].

Our study provides new information at two levels. Firstly, it showed that- although significant in the two groups- the extent of seasonal BPV was lower among CKD patients compared to patients without CKD. This finding did not reach statistical significance in our sample but needs to be further analyzed in large prospective studies that compare CKD patients with matched-controls. Secondly, our study provides interesting data on trends of antihypertensive medications' adjustment based on seasonal BPV. Indeed, it showed that antihypertensive dose reduction and removal occurred during the warm season whereas adding treatment was predominant in the cold season. The reduction of treatment during summer period was reported by a Japanese study where 13.5% of patients needed a decrease in treatment to avoid hypotension, mainly in diuretics [38]. This continuous adjustment of antihypertensive treatment among our patients maintained the SBP/DBP at a mean level of 138/77 mmHg while being treated with a mean of 3.3 antihypertensive drugs. It is true that the most recent guidelines recommended a target BP <130/80 mmHg [9] but this study included patients followed prior to these guidelines. Our study lacks as well data about home BP and it is very hard to conclude from the office BP measurements if our patients had a well-controlled BP despite the frequent therapeutical adjustment.

Moreover, this study demonstrated a significant association between all seasonal modifications of antihypertensive treatment and eGFR loss. This concurs well with the findings of Chia et al that evaluated 825 patients and concluded that long term BPV was an independent predictor of renal function deterioration [39]. In addition to eGFR loss, our study analyzed the dialysis as another renal outcome. It showed that increasing antihypertensive medications in both seasons was associated with dialysis occurrence. This is an expected outcome since severe hypertension induces CKD progression and advanced kidney disease is prevented by an intensive BP management [40].

Finally, and interestingly, drug dose reduction during the warm season was found to be an independent factor associated with a 2.5 higher risk of death in our sample of patients. It is less likely that treatment reduction affected death by-itself; this result would rather be a reflection of the cause behind the low BP that required a treatment modification. Low BP is strongly associated with a higher stroke mortality [41]. The low BP could also be explained by an orthostatic hypotension or a low cardiac output, both known to increase death risk [42–44]. On the other hand-even if it is less plausible among our patients- it is noteworthy that inappropriate downward titration of antihypertensive drugs on the basis of variations of office blood pressure measurement during the hot season can reduce the extension of BP control over 24 h, and contribute to the paradoxical increase in night-time BP levels reported during hot weather in some studies [17,45]. We did not collect any data about the ambulatory night-time BP in our

**Table 6.** **a. Univariate regression analysis of factors associated with dialysis.** b. Adjusted logistic regression analysis of factors associated with dialysis.

| Variable | OR | 95% Confidence Interval | P |
|---|---|---|---|
| **Age** | 1.02 | 1.00; 1.04 | 0.020 |
| **Sex,** *Ref: Male* | 0.52 | 0.29; 0.91 | 0.022 |
| **eGFR< 60 mL/min/1.73 m$^2$** | 26.41 | 6.38; 109.42 | <0.001 |
| **Diabetes** | 1.16 | 0.71; 1.89 | 0.564 |
| **Obesity** | 0.49 | 0.26; 0.94 | 0.033 |
| **Difference in Systolic Blood Pressure between cold and warm season** | 1.01 | 0.99; 1.03 | 0.276 |
| **Difference in Diastolic Blood Pressure between cold and warm season** | 1.02 | 0.99; 1.05 | 0.112 |
| **Average Systolic Blood Pressure during cold season** | 1.03 | 1.01; 1.05 | <0.001 |
| **Average Diastolic Blood Pressure during cold season** | 1.02 | 0.99; 1.04 | 0.246 |
| **Average Pulse Pressure during cold season** | 1.03 | 1.01; 1.04 | 0.003 |
| **Average Systolic Blood Pressure during warm season** | 1.02 | 1.00; 1.03 | 0.034 |
| **Average Diastolic Blood Pressure during warm season** | 0.99 | 0.96; 1.02 | 0.397 |
| **Average Pulse Pressure during warm season** | 1.02 | 1.01; 1.04 | 0.011 |
| **Number of antihypertensive treatments** | 1.06 | 0.88; 1.28 | 0.540 |
| **ACE inhibitors** | 1.04 | 0.59; 1.83 | 0.907 |
| **Angiotensin receptor blockers** | 0.74 | 0.43; 1.26 | 0.265 |
| **Calcium channel blockers** | 2.87 | 1.26; 6.52 | 0.012 |
| **Alpha-blockers** | 1.97 | 1.12; 3.46 | 0.019 |
| **Beta-blockers** | 1.33 | 0.74; 2.41 | 0.344 |
| **Thiazide diuretics** | 0.32 | 0.17; 0.59 | <0.001 |
| **Moxonidine** | 2.71 | 1.57; 4.66 | <0.001 |
| **Reduction of dose during warm season** | 1.68 | 0.95; 2.97 | 0.071 |
| **Removal of medication during warm season** | 1.49 | 0.89–2.51 | 0.132 |
| **Increase of dose during warm season** | 1.36 | 0.77; 2.41 | 0.283 |
| **Adding medication during warm season** | 1.67 | 1.01; 2.78 | 0.048 |
| **Change of medication during warm season** | 1.19 | 0.71; 1.98 | 0.508 |
| **Increase of dose during cold season** | 1.84 | 1.09; 3.08 | 0.022 |
| **Adding medication during cold season** | 1.24 | 0.75; 2.07 | 0.400 |
| **Reduction of dose during cold season** | 1.08 | 0.53; 2.16 | 0.840 |
| **Removal of medication during cold season** | 1.88 | 1.04; 3.39 | 0.037 |
| **Change of medication during cold season** | 0.94 | 0.55; 1.59 | 0.811 |

| | Adjusted to age, sex and baseline eGFR | | | Adjusted to age, sex, baseline eGFR, number of antihypertensive treatments, average systolic and diastolic BP | | |
|---|---|---|---|---|---|---|
| Variable | OR | 95% Confidence Interval | P | OR | 95% Confidence Interval | P |
| **Obesity** | 0.77 | 0.36; 1.63 | 0.497 | | | |
| **Average Systolic Blood Pressure during cold season** | 1.02 | 1.01; 1.04 | 0.014 | 1.05 | 1.00; 1.09 | 0.05 |
| **Average Pulse Pressure during cold season** | 1.02 | 0.99; 1.04 | 0.175 | | | |
| **Average Systolic Blood Pressure during warm season** | 1.01 | 0.99; 1.03 | 0.415 | | | |
| **Average Pulse Pressure during warm season** | 1.01 | 0.99; 1.03 | 0.284 | | | |
| **Calcium channel blockers** | 1.98 | 0.78; 5.02 | 0.151 | | | |
| **Alpha-blockers** | 1.01 | 0.52; 1.97 | 0.972 | | | |
| **Thiazide diuretics** | 0.55 | 0.27; 1.12 | 0.100 | | | |
| **Moxonidine** | 1.29 | 0.68; 2.44 | 0.439 | | | |
| **Adding medication during warm season** | 2.09 | 1.01; 3.85 | 0.018 | 2.39 | 1.16; 4.89 | 0.018 |
| **Increase of dose during cold season** | 2.26 | 1.21; 4.22 | 0.010 | 2.43 | 1.19; 4.99 | 0.015 |
| **Removal of medication during cold season** | 1.81 | 0.90; 3.63 | 0.096 | | | |

Note. OR, odds ratio.

**Table 7. a. Univariate regression analysis of factors associated with death.** b. Adjusted logistic regression analysis of factors associated with death.

| Variable | OR | 95% Confidence Interval | P |
|---|---|---|---|
| **Age** | 1.04 | 1.02; 1.07 | 0.001 |
| **Sex**, *Ref: Male* | 0.69 | 0.37; 1.28 | 0.242 |
| **eGFR< 60 mL/min/1.73 m$^2$** | 2.74 | 1.66; 4.49 | <0.001 |
| **Diabetes** | 1.29 | 0.74; 2.29 | 0.366 |
| **Obesity** | 0.53 | 0.26; 1.10 | 0.091 |
| **Difference in Systolic Blood Pressure between cold and warm season** | 1.01 | 0.99; 1.03 | 0.607 |
| **Difference in Diastolic Blood Pressure between cold and warm season** | 1.02 | 0.98; 1.05 | 0.365 |
| **Average Systolic Blood Pressure during cold season** | 1.01 | 0.99; 1.02 | 0.555 |
| **Average Diastolic Blood Pressure during cold season** | 0.98 | 0.96; 1.01 | 0.263 |
| **Average Pulse Pressure during cold season** | 1.01 | 0.99; 1.03 | 0.176 |
| **Average Systolic Blood Pressure during warm season** | 1.01 | 0.99; 1.02 | 0.585 |
| **Average Diastolic Blood Pressure during warm season** | 0.97 | 0.94; 0.99 | 0.027 |
| **Average Pulse Pressure during warm season** | 1.02 | 1.00; 1.04 | 0.048 |
| **Number of antihypertensive treatments** | 0.79 | 0.62; 1.02 | 0.076 |
| **ACE inhibitors** | 1.20 | 0.64; 2.25 | 0.571 |
| **Angiotensin receptor blockers** | 0.74 | 0.41; 1.36 | 0.335 |
| **Calcium channel blockers** | 0.78 | 0.39; 1.53 | 0.466 |
| **Alpha-blockers** | 0.68 | 0.32; 1.48 | 0.330 |
| **Beta-blockers** | 1.54 | 0.76; 3.10 | 0.229 |
| **Thiazide diuretics** | 0.37 | 0.19; 0.75 | 0.005 |
| **Moxonidine** | 0.95 | 0.47; 1.92 | 0.882 |
| **Reduction of dose during warm season** | 3.07 | 1.67; 5.64 | <0.001 |
| **Removal of medication during warm season** | 2.02 | 1.13–3.62 | 0.018 |
| **Increase of dose during warm season** | 1.66 | 0.89; 3.11 | 0.110 |
| **Adding medication during warm season** | 1.38 | 0.78; 2.45 | 0.275 |
| **Change of medication during warm season** | 1.41 | 0.79; 2.51 | 0.244 |
| **Increase of dose during cold season** | 0.99 | 0.55; 1.81 | 0.986 |
| **Adding medication during cold season** | 1.00 | 0.56; 1.78 | 0.992 |
| **Reduction of dose during cold season** | 1.09 | 0.49; 2.39 | 0.827 |
| **Removal of medication during cold season** | 1.05 | 0.51; 2.17 | 0.889 |
| **Change of medication during cold season** | 1.24 | 0.69; 2.23 | 0.472 |

| | Adjusted to age, sex and baseline eGFR | | | Adjusted to age, sex, baseline eGFR, number of antihypertensive treatments, average systolic and diastolic BP | | |
|---|---|---|---|---|---|---|
| **Variable** | OR | 95% Confidence Interval | P | OR | 95% Confidence Interval | P |
| **Average Diastolic Blood Pressure during warm season** | 0.99 | 0.95; 1.02 | 0.464 | | | |
| **Average Pulse Pressure during warm season** | 1.00 | 0.98; 1.02 | 0.849 | | | |
| **Thiazide diuretics** | 0.47 | 0.23; 0.97 | 0.042 | 0.74 | 0.29; 1.86 | 0.527 |
| **Reduction of dose during warm season** | 2.45 | 1.30; 4.62 | 0.005 | 2.89 | 1.36; 6.18 | 0.006 |
| **Removal of medication during warm season** | 1.69 | 0.93; 3.09 | 0.088 | | | |

Note. OR, odds ratio.

patients but it is unlikely that these patients would have a higher death risk than patients who required an increase in treatment during summer, even if they increased their night-time BP levels. Although Mallamaci et al showed that long-term systolic BPV rather than short-term BPV in CKD patients increases the risk of death and cardiovascular events [46], our study did

not show higher risk of death associated with the seasonal difference in SBP but rather with low diastolic and high pulse pressure in the warm season.

## Strengths and limitations

This study is the first to assess blood pressure variability in a Mediterranean climate. It is also one of the few assessing a difference between CKD and non-CKD patients among patients diagnosed with hypertension. Furthermore, it includes a representative sample size of the Lebanese hypertensive population. The limitations of this study are those related to its retrospective design and the lack of data on ambulatory blood pressure levels. In addition, the room temperature where the blood pressure was measured was not documented but we believe that it correlated with the weather temperature across seasons.

## Suggestions for the future

With the global warming and future extreme weather temperatures in both cold and hot seasons, it is essential for clinicians to acknowledge the seasonal BP in CKD patients and recognize patients who require a modification of their antihypertensive treatments. This is a call as well for scientists and experts to take into consideration within trials and guidelines the seasonal BPV in chronic kidney disease patients and establish recommendations to stratify patients based on their risks and recommend more seasonal visits to avoid over- or under-treatment.

## Conclusions

This study underscored the seasonal blood pressure variability in a Mediterranean climate. Blood pressure gets lower in warm seasons and higher in cold seasons in both CKD and non-CKD hypertensive patients. This blood pressure variability is related to a significant eGFR loss. The need to increase treatment in both seasons is associated with high risk of dialysis. A low BP during the warm season is also found highly associated with death. All these variations are necessary to recognize and acknowledge since they require treatment adjustments and might be important prognostic factors.

## Supporting information

**S1 Fig. Flow diagram of patients' inclusion.**
(DOCX)

**S1 Table. Average monthly temperatures between 2006 and 2020.**
(DOCX)

## Author Contributions

**Conceptualization:** Tatiana Charbel, Georgio El Koubayati, Mabel Aoun.

**Data curation:** Tatiana Charbel, Georgio El Koubayati, Chloe Kharsa, Mabel Aoun.

**Formal analysis:** Mabel Aoun.

**Investigation:** Tatiana Charbel, Mabel Aoun.

**Methodology:** Georgio El Koubayati, Mabel Aoun.

**Project administration:** Mabel Aoun.

**Resources:** Mabel Aoun.

**Supervision:** Mabel Aoun.

**Validation:** Mabel Aoun.

**Visualization:** Tatiana Charbel, Mabel Aoun.

**Writing – original draft:** Tatiana Charbel, Georgio El Koubayati, Chloe Kharsa.

**Writing – review & editing:** Tatiana Charbel, Georgio El Koubayati, Mabel Aoun.

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
