## [Decision Letter · Decision Letter 0]

5 Sep 2023

PONE-D-23-20485Acknowledging the Impact of Seasonal Blood Pressure Variation in Hypertensive CKD and non-CKD Patients Living in a Mediterranean ClimatePLOS ONE

Dear Dr. Aoun,

Thank you for submitting your manuscript to PLOS ONE. After careful consideration, we feel that it has merit but does not fully meet PLOS ONE’s publication criteria as it currently stands. Therefore, we invite you to submit a revised version of the manuscript that addresses the points raised during the review process.

We look forward to receiving your revised manuscript.

Kind regards,

Tauqeer Hussain Mallhi, Ph.D

Academic Editor

PLOS ONE

Additional Editor Comments:

Dear Authors, thank you for submitting in Plos One. Your manuscript has been assessed by relevant experts from the field. They found the manuscript interesting but raised several concerns in methodology, particularly analysis section, and interpretation of results. I believe that adjustment of comparisons is required for valid conclusions. It is requested to please consider the comments of reviewers.

Reviewers' comments:

Reviewer's Responses to Questions

**Comments to the Author**

1. Is the manuscript technically sound, and do the data support the conclusions?

Reviewer #1: Partly

Reviewer #2: Yes

Reviewer #3: Partly

2. Has the statistical analysis been performed appropriately and rigorously? 

Reviewer #1: Yes

Reviewer #2: Yes

Reviewer #3: No

3. Have the authors made all data underlying the findings in their manuscript fully available?

Reviewer #1: Yes

Reviewer #2: Yes

Reviewer #3: Yes

4. Is the manuscript presented in an intelligible fashion and written in standard English?

Reviewer #1: Yes

Reviewer #2: Yes

Reviewer #3: Yes

5. Review Comments to the Author

Reviewer #1: There are many things to consider while measuring blood pressure, most of them are mentioned by the author but some are missed (may be author forgot to mention) which are important aspect of BP measurement. like

1. Physical environment (outdoor or indoor) and also room temperature where BP was measured has impact on BP value, as per studies, with change in room temperature BP values also varies.

2. Psychological and mental Status like anxiety, mood affects BP values. and these things need to be looked into in order to give a definitive results if the study.

Thanks,

Reviewer #2: Page 12.

Indeed, it showed that antihypertensive dose reduction and molecules’ removal occurred during the warm season whereas adding treatment was predominant in the cold season.

Clarify which antihypertensive medication were implicated in this fluctuation.

Explain and clarify the use of the word “molecule”. It does not seem to fit.

Conclusions:

Blood pressure gets lower in warm seasons and higher in cold seasons in both CKD and non-CKD hypertensive patients.

Explain is this is due to hydration status.

Reviewer #3: In this manuscript the authors describe the impact of seasonal blood pressure variation (BPV) in patients with or without chronic kidney disease (CKD) living in a Mediterranean climate. Similar seasonal BPV has been described in hypertensive patients and also among patients with chronic kidney disease (CKD). The authors conclude that treatment adjustments related to seasonal BPV may have important prognostic implications for patients.

More detail related to the methods would improve the clarity of the manuscript especially with respect to the timing of data collection. Some specific comments are presented below:

Methods

1) Variables collected: “Data collection started in August 2022” please confirm that this means “chart review was started” not collection of data for the study as it appears patients were seen between Feb 2006 and April 2020 so presumably data from this latter period was used for the analyses.

2) It is necessary to describe the interval for BP recording (between 2006-2020) was a mean for the warm/cold periods calculated for each patient? or was data just collected over the latest consecutive warm/cold periods. As the follow-up period for patients was variable (51.1 ± 44.3 months), what was the interval between T1 and T2 did this differ between CKD and non-CKD patients?

3) Seasonal variation in blood pressure has be linked with vitamin D levels. It would be helpful to know and include information on 25-OH vitamin D levels in the studied patients if available.

4) No adjustment for multiple comparisons was included.

Results

5) Consider using medication or drug instead of molecule.

6) Table 1 follow up and outcomes: As the follow-up time was variable- data on eGFR change from baseline is difficult to interpret it may be better expressed as eGFR change/year.

7) Table 3: Comparison of patients with CKD vs. non-CKD.

Were the data collection periods for CKD and non-CKD patients the same. CKD progression over a longer period may affect BPV independent of season.

8) Table 4 Please describe in table footnote what “Change of molecule” includes. There does not appear to be any difference between warm/cold season with respect to change of molecule.

Discussion

9) Paragraph 1: The authors do not address seasonal variation in 25-OH vitamin D levels as a possible contributor to BPV.

10) Paragraph 2: The authors indicate that BPV was lower among CKD cases compared to those without CKD, however, it is unclear whether this difference was significant.

6. PLOS authors have the option to publish the peer review history of their article (what does this mean?). If published, this will include your full peer review and any attached files.

Reviewer #1: **Yes: **Tshering Namgay

Reviewer #2: No

Reviewer #3: No

---

## [Author Response · Author response to Decision Letter 0]

25 Sep 2023

Dear Editor,

Thank you for giving us the opportunity to submit a revised version of our manuscript.

Please find below a point-by-point response to reviewers' comments.

Editor's comment

Dear Authors, thank you for submitting in Plos One. Your manuscript has been assessed by relevant experts from the field. They found the manuscript interesting but raised several concerns in methodology, particularly analysis section, and interpretation of results. I believe that adjustment of comparisons is required for valid conclusions. It is requested to please consider the comments of reviewers.

Authors' response to Editor's comment:

Thank you for your valuable comments. We added details to the methodology and results that clarified several points raised by the reviewers. On the other hand, the editor and the reviewer #3 both raise an important point regarding the adjustment of comparisons, however given our sample size, the nature and focus of our study, we believe it is important to show all results given the limited data about this topic. Adjustment for multiple comparisons is not that simple as emphasized in the two references below. If the editor insists, we can still submit an edited version with multiple comparison adjustment.

1. Rothman KJ. No adjustments are needed for multiple comparisons. Epidemiology. 1990 Jan;1(1):43-6. PMID: 2081237.

2. Althouse A. D. (2016). Adjust for Multiple Comparisons? It's Not That Simple. The Annals of thoracic surgery, 101(5), 1644–1645. https://doi.org/10.1016/j.athoracsur.2015.11.024

Reviewer #1.

There are many things to consider while measuring blood pressure, most of them are mentioned by the author but some are missed (may be author forgot to mention) which are important aspect of BP measurement. like

1. Physical environment (outdoor or indoor) and also room temperature where BP was measured has impact on BP value, as per studies, with change in room temperature BP values also varies.

2. Psychological and mental Status like anxiety, mood affects BP values. and these things need to be looked into in order to give a definitive results if the study.

Thanks,

Authors' response to reviewer #1:

Thank you for your important comments. We agree with the reviewer that many factors are indeed important to consider while measuring BP and we thank the reviewer for giving us the chance to add the details that are usually followed during a BP measurement: like the seated position, after 10 minutes of rest, in a quiet environment. The room temperature and mood state might also affect the blood pressure levels. However, due to the retrospective design of the study, the room temperature was not documented in the charts and we believe that the indoor temperature blood pressure at the time of visit varied in the same direction as the weather temperature and no room was overheated or overcooled. We added this to the limitations' paragraph. Regarding the mood status we believe that this factor can only be addressed by measuring blood pressure in a quiet environment after some minutes of rest as recommended by the guidelines. 

Reviewer #2.

Very interesting article. Just a few observations. 

Authors' response: We are grateful for the reviewer's positive feedback.

Reviewer #2, comment #1:

Page 12.

Indeed, it showed that antihypertensive dose reduction and molecules’ removal occurred during the warm season whereas adding treatment was predominant in the cold season. 

Clarify which antihypertensive medication were implicated in this fluctuation.

Explain and clarify the use of the word “molecule”. It does not seem to fit.

Authors' response to Reviewer #2, comment #1: Thank you for this important comment. We added to the results the following: " Among the 82 reduced medications in warm seasons, 32 were RAASi (39%), 27 CCBs (32.9%), 21 diuretics (25.6%), 19 beta-blockers (23.2%). Among the removed 122 medications in warm seasons: 53 were RAASi (43.4%), 36 diuretics (29.5%), 34 CCB (27.9%)." and the following: "Among the 131 increased medications, 53 (40.5%) were CCBs, 41 (31.3%) diuretics, 33 (25.2%) RAASi, 7 (5.3%) beta-blockers, 5 (3.8%) alpha-blockers and 4 (3%) moxonidine. Among the added 182 medications during cold season: 62 were RAASi (34.1%), 46 diuretics (25.3%), 46 CCBs (25.3%), 16 beta-blockers (8.8%), 15 moxonidine (8.2%), 13 alpha-blockers (7.1%)." 

We replaced "molecule" by "medication" throughout the manuscript.

Reviewer #2 comment #2:

Conclusions:

Blood pressure gets lower in warm seasons and higher in cold seasons in both CKD and non-CKD hypertensive patients.

Explain is this is due to hydration status.

Authors' response to Reviewer #2, comment #2:

Thanks for this comment. Hydration status might be one of the factors associated with this BP variation. Based on the first comment of the reviewer, we can see that diuretics are involved in one third of cases of treatment modifications. We explained the reason behind this seasonal variation in the discussion paragraph: " This can be explained by physiological thermoregulation like temperature-induced vasoconstriction and vasodilation, climate-related perspiration, changes in dietary and water intake [36]". We added " Our results point out as well to a change in hydration status knowing that 26% of patients stopped diuretics and 30% of them removed diuretics during the warm season."

Reviewer #3.

In this manuscript the authors describe the impact of seasonal blood pressure variation (BPV) in patients with or without chronic kidney disease (CKD) living in a Mediterranean climate. Similar seasonal BPV has been described in hypertensive patients and also among patients with chronic kidney disease (CKD). The authors conclude that treatment adjustments related to seasonal BPV may have important prognostic implications for patients.

More detail related to the methods would improve the clarity of the manuscript especially with respect to the timing of data collection. Some specific comments are presented below:

Reviewer #3, comment #1:

Methods

1) Variables collected: “Data collection started in August 2022” please confirm that this means “chart review was started” not collection of data for the study as it appears patients were seen between Feb 2006 and April 2020 so presumably data from this latter period was used for the analyses.

Authors' response to Reviewer #3, comment #1:

Thank you for your comment. We clarified that collection through chart review started in August 2022.

Reviewer #3, comment #2:

2) It is necessary to describe the interval for BP recording (between 2006-2020) was a mean for the warm/cold periods calculated for each patient? or was data just collected over the latest consecutive warm/cold periods. 

Authors' response to Reviewer #3, comment #2: Thank you for your comment. We added to the Methods the following paragraph: 

"Collection and averaging of BP levels

SBP and DBP levels were collected for every patient from all their visits across all seasons, before any treatment adjustments were made during each visit. These SBP and DBP levels were grouped and averaged for each cold or warm season, for each patient."

Reviewer #3, comment #3:

As the follow-up period for patients was variable (51.1 ± 44.3 months), what was the interval between T1 and T2 did this differ between CKD and non-CKD patients?

Authors' response to Reviewer #3, comment #3:

This is an interesting question. The interval between T1 (first visit) and T2 (last visit) is the duration of follow-up of patients. It was 49.9±42.5 months in CKD patients and 57.2 ±53 months in non-CKD patients (the difference did not reach a significant level, P=0.319). We added the means and SDs to the results.

Reviewer #3, comment #4:

3) Seasonal variation in blood pressure has been linked with vitamin D levels. It would be helpful to know and include information on 25-OH vitamin D levels in the studied patients if available.

Authors' response to Reviewer #3, comment #4:

Unfortunately, we did not collect the 25OH Vitamin D levels of patients. This requires as well a follow-up with several levels across several seasons (as we did for the BP) to show any association and we did not have such information. 

Reviewer #3, comment #5:

4) No adjustment for multiple comparisons was included.

Authors' response to Reviewer #3, comment #5:

As mentioned earlier, the editor and the reviewer #3 both raise an important point regarding the adjustment of comparisons, however given our sample size, the nature and focus of our study, we believe it is important to show all results given the limited data about this topic. Adjustment for multiple comparisons is not that simple as emphasized in the two references below. If the editor insists, we can still submit an edited version with multiple comparison adjustment.

1. Rothman KJ. No adjustments are needed for multiple comparisons. Epidemiology. 1990 Jan;1(1):43-6. PMID: 2081237.

2. Althouse A. D. (2016). Adjust for Multiple Comparisons? It's Not That Simple. The Annals of thoracic surgery, 101(5), 1644–1645. https://doi.org/10.1016/j.athoracsur.2015.11.024

Reviewer #3, comment #6:

Results

5) Consider using medication or drug instead of molecule.

Authors' response to Reviewer #3, comment #6: done.

Reviewer #3, comment #7:

6) Table 1 follow up and outcomes: As the follow-up time was variable- data on eGFR change from baseline is difficult to interpret it may be better expressed as eGFR change/year.

Authors' response to Reviewer #3, comment #7:

Thank you for this important comment. We added a row to Table 1 that includes the median of eGFR change per year.

Reviewer #3, comment #8:

7) Table 3: Comparison of patients with CKD vs. non-CKD.

Were the data collection periods for CKD and non-CKD patients the same. CKD progression over a longer period may affect BPV independent of season.

Authors' response to Reviewer #3, comment #8: Based on the reviewer's suggestion, we analyzed the duration of follow-up and found it shorter in CKD patients (comment #3).

Reviewer #3, comment #9:

8) Table 4 Please describe in table footnote what “Change of molecule” includes. There does not appear to be any difference between warm/cold season with respect to change of molecule.

Authors' response to Reviewer #3, comment #9: Thank you for this comment. Change of molecule or medication means replacing one medication by another with a known equivalent dose (meaning no increasing or decreasing number or doses of medications). We added this clarification to the footnote of Table 4. 

Reviewer #3, comment #10:

Discussion

9) Paragraph 1: The authors do not address seasonal variation in 25-OH vitamin D levels as a possible contributor to BPV.

Authors' response to Reviewer #3, comment #10:

Thank you for this comment and for giving us the opportunity to make our discussion more comprehensive. We added the following to the discussion: 

"An additional reason behind the lower BP in warm seasons could be related to vitamin D [37]. Although interventional trials did not show a decrease in BP following vitamin D supplementation, several studies emphasized the relationship between exposure to sunshine and the release of nitric oxide from the skin [37]."

We added this reference [37] "Adamczak M, Surma S, Więcek A. Vitamin D and Arterial Hypertension: Facts and Myths. Curr Hypertens Rep. 2020;22(8):57. Published 2020 Jul 15. doi:10.1007/s11906-020-01059-9", unless the reviewer has a better reference to suggest.

Reviewer #3, comment #11:

10) Paragraph 2: The authors indicate that BPV was lower among CKD cases compared to those without CKD, however, it is unclear whether this difference was significant.

Authors' response to Reviewer #3, comment #11:

The reviewer raises a relevant point. This difference was indeed not statistically significant but it was consistent for systolic and diastolic blood pressure. This difference could potentially appear significant in larger studies and we wanted to highlight this point to open the door for future studies. We amended the statement in the discussion as follows: " Firstly, it showed that- although significant in the two groups- the extent of seasonal BPV was lower among CKD patients compared to patients without CKD. This finding did not reach statistical significance in our sample but needs to be further analyzed in large prospective studies that compare CKD patients with matched-controls."

---

## [Decision Letter · Decision Letter 1]

12 Oct 2023

Acknowledging the Impact of Seasonal Blood Pressure Variation in Hypertensive CKD and non-CKD Patients Living in a Mediterranean Climate

PONE-D-23-20485R1

Dear Dr. Aoun,

We’re pleased to inform you that your manuscript has been judged scientifically suitable for publication and will be formally accepted for publication once it meets all outstanding technical requirements.

Kind regards,

Tauqeer Hussain Mallhi, Ph.D

Academic Editor

PLOS ONE

Additional Editor Comments (optional):

Reviewers' comments:

Reviewer's Responses to Questions

**Comments to the Author**

1. If the authors have adequately addressed your comments raised in a previous round of review and you feel that this manuscript is now acceptable for publication, you may indicate that here to bypass the “Comments to the Author” section, enter your conflict of interest statement in the “Confidential to Editor” section, and submit your "Accept" recommendation.

Reviewer #1: All comments have been addressed

Reviewer #3: All comments have been addressed

2. Is the manuscript technically sound, and do the data support the conclusions?

Reviewer #1: Yes

Reviewer #3: (No Response)

3. Has the statistical analysis been performed appropriately and rigorously? 

Reviewer #1: Yes

Reviewer #3: Yes

4. Have the authors made all data underlying the findings in their manuscript fully available?

Reviewer #1: Yes

Reviewer #3: Yes

5. Is the manuscript presented in an intelligible fashion and written in standard English?

Reviewer #1: Yes

Reviewer #3: Yes

6. Review Comments to the Author

Reviewer #1: since all the things has been included in this revision, i would say this study will have a great impact on the treatment module for Hypertensive patients.

thanks

Reviewer #3: The authors have added additional description to the methods significantly improving clarity of the manuscript. Limitations are now more fully discussed. I have no further comments.

7. PLOS authors have the option to publish the peer review history of their article (what does this mean?). If published, this will include your full peer review and any attached files.

Reviewer #1: **Yes: **Tshering Namgay

Reviewer #3: No

---

## [Editor Report · Acceptance letter]

16 Oct 2023

PONE-D-23-20485R1 

Acknowledging the Impact of Seasonal Blood Pressure Variation in Hypertensive CKD and non-CKD Patients Living in a Mediterranean Climate 

Dear Dr. Aoun:

I'm pleased to inform you that your manuscript has been deemed suitable for publication in PLOS ONE. Congratulations! Your manuscript is now with our production department. 

Kind regards, 

on behalf of

Dr. Tauqeer Hussain Mallhi 

Academic Editor

PLOS ONE